# Heat Shock Related Protein Expression in Abdominal Testes of Asian Elephant (*Elephas maximus*)

**DOI:** 10.3390/ani14152211

**Published:** 2024-07-30

**Authors:** Yoko Sato, Theerawat Tharasanit, Chatchote Thitaram, Chaleamchat Somgird, Sittidet Mahasawangkul, Nikorn Thongtip, Kaywalee Chatdarong, Narong Tiptanavattana, Masayasu Taniguchi, Takeshige Otoi, Mongkol Techakumphu

**Affiliations:** 1Department of Biology, School of Biological Sciences, Tokai University, Sapporo 0058601, Japan; 2Department of Obstetrics, Gynaecology and Reproduction, Faculty of Veterinary Science, Chulalongkorn University, Bangkok 10330, Thailand; theerawat.t@chula.ac.th (T.T.); kaywalee.c@chula.ac.th (K.C.); mongkol.t@chula.ac.th (M.T.); 3Veterinary Clinical Stem Cells and Bioengineering Research Unit, Chulalongkorn University, Bangkok 10330, Thailand; 4Center of Elephant and Wildlife Health, Faculty of Veterinary Medicine, Chiang Mai University, Chiang Mai 50100, Thailand; chatchote.thitaram@cmu.ac.th (C.T.); chaleamchat.s@cmu.ac.th (C.S.); 5The Royal Initiated Thai Elephant Conservation Center, Lampang 52190, Thailand; msittidet@hotmail.com; 6Faculty of Veterinary Medicine, Kasetsart University, Nakhon Pathom 73140, Thailand; nthongtip@yahoo.com; 7Faculty of Veterinary Science, Prince of Songkla University, Songkhla 90110, Thailand; narong.ti@psu.ac.th; 8Department of Animal Reproduction, Joint Faculty of Veterinary Medicine, Yamaguchi University, Yamaguchi 7538515, Japan; masa0810@yamaguchi-u.ac.jp; 9Bio-Innovation Research Center, Tokushima University, Tokushima 7793233, Japan; otoi@tokushima-u.ac.jp

**Keywords:** abdominal testis, spermatogenesis, elephant, heat shock factor

## Abstract

**Simple Summary:**

In the abdominal testes of most mammals, with the exception of elephants and a few animal species, spermatogenesis is characterised by an arrest of spermatogenesis due to heat stress. However, the mechanisms underlying normal spermatogenesis in the elephant abdominal testis remain unknown. We hypothesised that comparing the expression of heat stress tolerance molecules of elephant testes with that of other animals and investigating the mechanism of spermatogenesis in elephants might provide clues to overcome the causes of heat stress-induced spermatogenesis dysfunction. The results showed that the elephant’s body cells, other than the testes, have a similar mechanism to those of other mammals but that the spermatogenic cells have a different mechanism, with different molecular immunoexpression under heat stress in the abdomen. These findings suggest that these molecules and their relationships may be important in future investigations of the mechanisms that overcome heat stress to maintain spermatogenesis in elephants.

**Abstract:**

The abdominal testes of Asian elephants show normal spermatogenesis. Heat shock in cryptorchid testes elevates heat shock factor (HSF) expression, leading to germ cell apoptosis, while increased heat shock proteins (HSPs) levels provide protection. To investigate how heat shock affects elephant spermatogenic cells, focusing on heat shock-related molecules and the cell death mechanism, immunohistochemistry and TUNEL staining were employed to assess the immunoexpression of several heat shock-related molecules and the status of apoptosis in elephant fibroblasts (EF) induced by heat shock stimulus. Additionally, the immunoexpression of heat shock-related molecules and cell proliferation status in the elephant spermatogenic cells. Our finding indicated that heat shock-induced HSF1 immunoexpression in EF leads to apoptosis mediated by T-cell death-associated gene 51 (TDAG51) while also upregulating HSP70 to protect damaged cells. In elephant spermatogenic cells, immunostaining revealed a predominance of proliferating cell nuclear antigen (PCNA)-positive cells with minimal TDAG51- and TUNEL-positive cells, suggesting active proliferation and apoptosis suppression during normal spermatogenesis in the abdominal testis. Interestingly, spermatogonia co-immunoexpressed HSF1 and HSP90, potentially reducing apoptosis through protective mechanisms different from those observed in other mammals. Spermatogenic cells did not show immunolocalisation of HSP70, and hence, it may not contribute to protecting the spermatogonia from heat shock because the transcriptional activity of HSF1 is suppressed by HSP90A binding. This study provides insight into the specific heat shock response and defence mechanisms in elephant spermatogenic cells and may contribute to our understanding of species-specific adaptation to environmental stresses of the testis.

## 1. Introduction

Testes in most mammals are maintained at 2–8 °C below the body temperature (as they are located in the scrotum, which is outside the body cavity) through a counter-current heat-exchange system that cools the blood entering the testes [1,2]. Abdominal testes in most mammals, except for elephants and dolphins, are characterised by spermatogenic arrest due to heat-induced stress [3,4]. In dolphins, counter-current heat-exchange systems exist near the testes that provide cooler conditions [5]. However, in elephants, the testes display normal spermatogenesis in the abdomen without a counter-current heat-exchange system to cool the testes [6,7]. Therefore, the mechanisms underlying normal spermatogenesis in the abdominal testes of elephants remain unclear.

Spermatogenesis is easily disrupted by modest increases in temperature or by other adverse environmental factors, such as therapeutic drugs, food additives, industrial chemicals, solvents, and agrochemicals [3,8]. Stress induced by heat shock stimuli in cryptorchid testes increases the expression of heat shock factors (HSFs) and eliminates germ cells through apoptosis via the expression of T-cell death-associated gene 51 (TDAG51) [9,10]. In contrast, there is a mechanism that protects germ cells from heat shock stimuli through an increase in heat shock protein (HSP) expression, which assists in protein folding and inhibits protein denaturation [11,12]. HSF1 protects against cell death resulting from various types of stress, effects that are mediated by the regulation of HSPs [13,14] and other unidentified genes [15]. Thus, HSF1 may promote spermatogenic cell elimination and protection from apoptosis, which are mechanisms that depend on the equilibrium of heat stress-related gene expression. Alterations in this equilibrium can induce spermatogenesis arrest in the testes. Therefore, an appropriate molecular response to heat shock stimulus could rescue spermatogenic cells after heat shock.

Because elephant testes display normal spermatogenesis in the abdomen, we hypothesised that the response of elephant spermatogenic cells to heat shock stimuli, seen as a change in heat shock-related molecule expression, is different from that of other mammalian species. Accordingly, we examined the immunoexpression levels of heat shock-related molecules, such as HSF1, HSP70, and TDAG51, and apoptosis after heat shock stimulus using elephant fibroblast cells. In addition, immunohistochemistry and Terminal deoxynucleotidyl transferase (TdT) dUTP nick end labelling (TUNEL) were performed to detect heat shock-related molecules, apoptosis, and cell proliferation in the tissues of elephant testis.

## 2. Materials and Methods

### 2.1. Elephant Fibroblast Cells

Primary elephant fibroblasts were derived from infant ear skin. Cells (passage 9–11) were seeded on a 1% gelatin/phosphate-buffered saline (PBS)-coated round cover glass placed in a Falcon^R^ centre-well organ culture dish (Becton Dickinson, Franklin Lakes, NJ, USA) with complete culture medium. The culture medium had the following composition: Dulbecco’s Modified Eagle’s Medium (DMEM) (Gibco Life Technologies, Grand Island, NY, USA) with high glutamine (Gibco Life Technologies, Grand Island, NY, USA) supplemented with 20% foetal bovine serum (FBS) (Gibco Life Technologies, Grand Island, NY, USA), and 1% antibiotic-antimycotic (Thermo Fisher Scientific, Waltham, MA, USA). Cells were cultured and incubated at 37 °C and 5% CO_2_ in a humidified incubator. Cells showing 70% confluence after the culture period were used for the heat shock experiments.

### 2.2. Tissues of Elephant Testis

Asian elephant (*Elephas maximus*) testes were collected from four deceased adult bulls and two calves from the Thai Elephant Conservation Center and elsewhere in northern Thailand. To distinguish the adult and calf, approximately age determination was performed by checking the ear fold, teeth, as well as the wrinkle of the face and skin, and the concave of the skull by experts of the elephant [16,17]. In addition, in the case of the captive elephants, the information was also obtained by asking the mahouts and the record. The necropsy of the elephant followed the protocol [18]. The testes were removed from the abdominal cavity, as obviously located just behind the kidney hanging on the dorsal parietal wall of the abdomen [19]. For the younger elephant, the location and testis structure were the same, but the size was smaller. The testicles were maintained under 4 °C and moved to Chulalongkorn University under 4 °C. The testis was dissected, and the tissue of the testis was fixed with 4% paraformaldehyde (PFA) (VWR BDH Prolabo, Poole, UK) in PBS (PFA/PBS) for 48 h. The fixed tissues were rinsed with PBS three times for 1 h, dehydrated with serial alcohol concentrations (70–100%) for 1 h each, and then replaced with xylene three times for 1 h. Subsequently, the tissue samples were impregnated with paraffin wax three times for 1 h before routine embedding. The paraffin (4 µM thick) sections were prepared for histological examination. The spermatogenic status of these samples was confirmed by observation of the spermatogenic stage after haematoxylin and eosin staining.

This study was ethically performed and approved by The Animal Care and Use Protocol of the Faculty of Veterinary Science at Chulalongkorn University (Accession No. 1831028).

### 2.3. Heat Shock Experiment

Culture dishes containing elephant fibroblast cells were sealed with parafilm and incubated in a water bath (45 °C) for 45 min, which served as a heat shock stimulus [20]. For the control experiments (no heat shock), culture dishes containing cells in the culture medium were sealed with parafilm and incubated at 37 °C for 45 min. After the heat shock (or no heat shock) stimulus to the cells and the removal of parafilm seal from the dish, the culture dishes were moved to the incubator and re-cultured for different periods (1 h–24 h after stimulus) or the culture dishes were washed three times with PBS (pH 7.2) (Wako, Saitama, Japan) and fixed with 4% PFA/PBS for 15 min (0 h after heat shock). After the completion of the incubation period post the heat shock (or no heat shock) stimulus, the culture dishes with cells were rinsed three times with PBS and then fixed with 4% PFA/PBS for 15 min (1 h–24 h after heat shock).

### 2.4. Immunocytochemistry and Immunohistochemistry

Immunocytochemistry and immunohistochemistry were performed according to the previous methods, slightly modified following [21,22]. After fixation, coverslips containing cells were washed several times with PBS and then incubated with 0.05% Tween/PBS for 15 min to permeabilise the cells. Coverslips were washed with PBS and incubated with 3% H_2_O_2_-MeOH for 10 min to block endogenous peroxidase activity. Subsequently, the coverslips were pre-incubated with 10% goat serum (Sigma-Aldrich, St. Louis, MO, USA) in PBS for 1 h. Primary antibodies against HSF1 (1:100, 4356, rabbit polyclonal, Cell Signaling Tech, Danvers, MA, USA), HSP70 (1:100, C92F3A-5, mouse monoclonal, Calbiochem, San Diego, CA, USA), TDAG51 (1:50, RN-6E2, mouse monoclonal, Santa Cruz Biotech, Dallas, TX, USA), and β-actin (1:200, AC-14, mouse monoclonal, Sigma-Aldrich, St. Louis, MO, USA) were diluted in 10% BSA/PBS, added to the coverslips in tissue culture dishes and incubated for 2 h at 37 °C in a humidified chamber. Rabbit IgG (Dako, Tokyo, Japan) and mouse IgG antibodies (Millipore, Tokyo, Japan) were adjusted to the working concentration of each primary antibody and used as negative controls. After washing several times with PBS, the cover glasses were incubated with anti-rabbit IgG (Fab’) labelled with an amino acid polymer-peroxidase complex (Histofine Simple Stain MAX PO (R); Nichirei Co., Tokyo, Japan) or anti-mouse IgG (Fab’) labelled with an amino acid polymer-peroxidase complex (Histofine Simple Stain MAX PO (M); Nichirei Co., Tokyo, Japan), for 1 h at 37 °C in a humidified chamber. The coverslips were then washed, and the peroxidase complex was detected by incubation with amino ethyl carbazole (AEC, Histofine Simple Stain AEC Solution; Nichirei Co., Tokyo, Japan) according to the manufacturer’s instructions. The coverslips were counterstained with haematoxylin, except for samples stained with HSF1, which were used for image analysis, mounted using an aqueous permanent mounting solution (Nichirei Co., Tokyo, Japan) with a coverslip, and observed under a microscope.

Fixed tissues were embedded in paraffin, and the paraffin (4 µM thick) sections were prepared. The de-paraffinised and rehydrated paraffin sections were microwaved for 30 min in 10 mM citrate buffer (pH 6.0) for antigen retrieval. After washing the slides with distilled water, the sections were incubated with 3% H_2_O_2_-MeOH for 10 min to block endogenous peroxidase activity. After this process, immunohistochemistry was performed with antibodies against HSF1, HSP70, TDAG51, HSP90A (1:200, PA3-013, rabbit polyclonal, Thermo Fisher Scientific, Waltham, MA, USA), HSP60 (1:200, MA-33012, mouse monoclonal, Fisher Scientific, Roskilde, Denmark), PCNA (1:100, SNCL-PCNA, mouse monoclonal, Novocastra^TM^, Newcastle, UK), and β-actin using the same methods described above for immunocytochemistry experiments, but with some variations: incubation with the primary antibody was performed overnight, and all incubations were performed at 25 °C.

### 2.5. TUNEL Assay

To identify apoptotic germ cells, TUNEL staining was performed [23,24]. Paraffin (4 µM thick) sections from fixed tissues of each specimen were de-paraffinised and rehydrated. The sections were then digested with 100 µg/mL proteinase K (Wako Co. Ltd., Tokyo, Japan) in PBS at 37 °C for 15 min, rinsed several times in PBS, and pre-incubated with terminal deoxynucleotidyl transferase (TdT) (Roche Diagnostics, Tokyo, Japan) buffer in a humidified chamber for 30 min at 37 °C. For the TUNEL assay, after fixation, cells were washed several times with PBS (−) and incubated with 0.1% Triton/0.1% sodium citrate for 15 min to permeabilise them. Cells were then rinsed several times with PBS and pre-incubated with TdT buffer following the same protocol as for TUNEL staining for tissue sections. After incubation, the sections and cells were reacted with 800 units/mL of TdT dissolved in TdT buffer supplemented with 0.1 µM biotin-16-dUTP (Roche Diagnostics, Tokyo, Japan), 20 µM dATP, 1.5 mM CoCl_2_, and 0.1 mM dithiothreitol at 37 °C in a humidified chamber for 90 min. As a negative control, the TUNEL reaction was performed without TdT. After several washings with distilled water and PBS, the sample signal was detected by incubation with peroxidase-labelled streptavidin (Nichirei Co., Tokyo, Japan) for 30 min at RT, followed by incubation with 3,3′ Diaminobenzidine (DAB, Histofine simple Stain DAB Solution; Nichirei Co., Tokyo, Japan) according to the manufacturer’s instructions. The sections and cells were counterstained with haematoxylin and mounted using the permanent Eukitt embedding medium (Eukitt quick-hardening mounting medium; Fluka Analytical, Waltham, MA, USA) for observation under a microscope.

### 2.6. Semi-Quantitative Image Analysis of Immunolabeling Cells

All images were acquired using a microscope (BX5, Olympus, Tokyo, Japan) and recorded using the DP2-BSW program (Olympus, Tokyo, Japan) and analysed using NIS imaging software (NIS Elements, ve.4.3, Nikon, Tokyo, Japan). The surface area occupied by the HSF1, HSP70, and β-actin immune-stained cells was measured in terms of optical density using NIS imaging software. After automatic binarisation processing of the images, the optical density data were reversed to adjust the luminance intensity. Finally, the staining intensity was measured and represented as arbitrary units [22]. Background intensities of the samples were quantified using the same method. The relative intensity of each antibody in the cells was calculated using the following formula:Relative intensity in cells = mean intensity of each antibody in cells − mean intensity of the background

The average arbitrary units of cells labelled with each antibody (per experiment) were determined based on the average of five fields (to determine the value of one field, we examined at least 10 points around the selected field and determined the average value as the field value) from each sample, with the exception of samples from TDAG51 immunostaining and TUNEL staining. For the TDAG51 or TUNEL staining of cells, we set a lower threshold value for staining intensity. Cells showing an intensity higher than the threshold were counted as positive. The data of TDAG51 or TUNEL staining in tissue samples are presented as the mean value of immune or TUNEL-positive cells/total cells in each field. Both TDAG51- and TUNEL-positive cells were manually counted in each test. The average of positive cells labelled with TDAG51 antibody or TUNEL (per experiment) was determined based on the average of five fields (to determine the value of one field, we examined at least 10 points around the selected field and determined the average value as the field value) from each sample. These examined fields were chosen randomly by moving slides with the stage under lower magnification by examiners blind to the tissue treatment. The data are presented as the mean percentage of immune-positive cells/total cells in five fields (immunocytochemistry).

### 2.7. Statistical Analysis

All values are expressed as mean ± SEs. Shapiro–Wilk normality test, one-way analysis of variance test, and Tukey-Kramer multiple comparison test as a post hoc analysis were performed using GraphPad 10 software (GraphPad Software, Boston, MA, USA). Differences with a probability value (*p*) of 0.05 or less were considered statistically significant.

## 3. Results

### 3.1. Effects of Heat Shock Stimulus on Elephant Fibroblasts

Elephant fibroblasts showed immunolocalisation of HSF1, HSP70, TDAG51, and β-actin (Figure 1, Figure 2, Figure 3 and Figure 4). Before the heat shock stimulus, the cells slightly immunoexpressed HSF1 (Figure 1A and Figure 5A). The immunoexpression of HSF1 in cells began to increase after heat shock stimulus at 0 h (Figure 1B and Figure 5A). At 1–2 h after the heat shock stimulus, the maximum immunoexpression of HSF1 in cells was significantly higher than that in cells without the heat shock stimulus (*p* < 0.05) (Figure 1C,D and Figure 5A). The immunoexpression gradually decreased until 6 h after the heat shock (Figure 1E,F and Figure 5A). Negative control did not show any immunostaining (Figure 1G).

HSP70 immunoexpression was undetectable before the heat shock stimulation (Figure 2A and Figure 5B). HSP70 immunolocalisation was observed in the nucleus 2 h after the heat shock stimulus (Figure 2D,J upper), and it expanded from the nucleus to the cytoplasm 4 h after the heat shock stimulus (Figure 2E,J middle). Finally, cells showed maximum immunoexpression of HSP70 in the cytoplasm at 6 h and 12 h after the heat shock stimulus (*p* < 0.05) (Figure 2F,G,J lower and Figure 5B) and gradually decreased its immunoexpression 18 h after the heat shock stimulus (Figure 2H,I and Figure 5B). Negative control did not show any immunostaining of HSP70 (Figure 2K).

In contrast, the immunoexpression of TDAG51 at 1–4 h after the heat shock stimulation was significantly higher than that in cells without the heat shock stimulation (*p* < 0.05) (Figure 3 and Figure 5C). Negative control did not show any immunostaining of TDAG51 (Figure 3J). Apoptosis was induced at 2 h after the heat shock in the elephant fibroblasts (Figure 5D). After an increase in the immunoexpression of HSP70 in the cells, TDAG51 immunoexpression and apoptosis were suppressed (Figure 5B–D).

β-actin immunoexpression levels in cells after the heat shock stimulus were almost the same as those in cells without the heat shock stimulus (Figure 4 and Figure 5E). Negative control did not show any immunostaining of β-actin (Figure 4J).

Furthermore, in cultures without the heat shock stimulus, all immunoexpression levels of HSF1, HSP70, TDAG51, TUNEL and β-actin in elephant fibroblast cells did not change (Table 1). These results suggest that the induction of HSF1, HSP70, TDAG51, and apoptosis in elephant fibroblasts was due to the heat shock stimulus without any effects from the culture conditions.

### 3.2. Heat Shock Related Molecules Immunoexpression in the Tissues of Elephant Testes

The molecules such as HSF1, TDAG51, and β-actin, which were observed in the elephant fibroblasts under heat shock stimulus, were also immunoexpressed in the cells of elephant abdominal testis, except HSP70 (Figure 6A,B,E,H). HSP90A was immunolocalized mainly in spermatogonia and Leydig cells, and HSP60 immunolocalization was found at almost all the cells in the testis (Figure 6C,D). Few TUNEL positive spermatogenic cells were observed, although there were many PCNA immunopositive spermatogonia (Figure 6F,G). Negative control did not show any immunostaining (Figure 6 insets).

## 4. Discussion

In the present study, HSF1 immunoexpression levels increased after heat shock stimulus in the elephant fibroblasts, suggesting that the response of HSF1 to heat shock stimulus is conserved in elephants, a phenomenon consistent with other mammals, such as mice [25].

HSF1 plays two roles in the mammalian testes: protection and elimination of damaged germ cells [9,26]. It has been suggested that *Tdag51*, a target gene of HSF1, subsequently activates apoptosis [9,10]. Increased immunoexpression of TDAG51 in elephant fibroblasts was observed after an increase in HSF1 immunoexpression, suggesting that the response of TDAG51 through HSF1 to the heat shock stimulus is conserved in elephant fibroblasts and is consistent with that in the other mammals such as mouse [9], rat [10]. Stress induced by heat shock stimuli in cryptorchid testes increases the expression of HSFs and eliminates germ cells through apoptosis via the expression of TDAG51 [9,10]. This suggests that, in most mammals, the inhibition of spermatogenesis during cryptorchidism is partly attributable to the activation of an HSF1/TDAG51-dependent mechanism. Normal spermatogenesis occurs in elephants even though their testes are located in the abdominal cavity and may be subjected to heat stress. The present study demonstrated the immunoexpression of HSF1 and TDAG51 in the spermatogenic cells of elephants, although the numbers of TDAG51 immunolocalised cells were quite smaller than the HSF1 immunolocalised cells. Furthermore, the number of TUNEL-positive cells was extremely low. TDAG51 immunoexpression in elephant spermatogenic cells in tissues under heat stress was lower than that in fibroblasts, suggesting that there is a protective mechanism against heat shock in testicular germ cells.

To protect somatic cells, HSF1 regulates the expression of the heat-inducible HSP70 form, which has anti-apoptotic functions [27]. In the present study, the elephant fibroblasts immunoexpressed HSP70 after heat shock stimulation, which is consistent with the other animal cell lines. In contrast, the germ cells of elephant testes did not immunoexpress HSP70 under heat stress in the abdomen, suggesting that the heat shock-stimulated HSF1 immunoexpression was not able to induce the immunoexpression of HSP70 in any cells of the elephant testis. In contrast, the present study showed that HSP90A is immunolocalised in the spermatogonia, whereas HSF1 is immunoexpressed in the elephant cells in the testis. Since the transcriptional activity of HSF1 is suppressed by the binding of HSP90A [28], the co-immunoexpression of HSF1 and HSP90A may suppress apoptosis in spermatogonia and may not require HSP70 immunoexpression to rescue the spermatogonia from heat shock. Furthermore, HSP90A promotes cancer cell proliferation [29]. We detected PCNA immunoexpression in the spermatogonia in the seminiferous tubules of elephant tissues in the testes. Reduced apoptosis and increased proliferation of spermatogonia may enable spermatogenesis in the elephant testes under heat stress in the abdominal cavity. Furthermore, HSP60 immunoexpression in almost all cells of elephant testis supports mitochondrial protein folding under heat stress in the abdominal cavity.

In the present study, we used heat shock in our experimental design to determine the ability of elephant fibroblast cells to respond to this kind of stimulus. The elephant’s body temperature is approximately 36 °C, with temperature variations between 35 and 37.5 °C [30,31]. Compared to the body temperature of other homeothermic mammals (Koala: 35.5–36.8 °C, human: 36–37 °C, mouse: 37–38 °C, horse: 37.5 °C, bovine: 38.5 °C, porcine: 38.9 °C, sheep: 39 °C), the body temperature of the elephant is low. In normal spermatogenesis, the testes of most mammals are maintained at 2–8 °C below body temperature [1]. However, there is no evidence whether there is a common appropriate temperature for spermatogenesis in mammals. For example, in humans, the scrotal temperature is around 34 °C in healthy men [32]. However, the scrotal temperature of a man with varicocele is 35.5 °C, and oligozoospermia is a common finding [33]. In humans, an increase in testicular temperature of 1 °C results in a decrease in normal sperm production [34]. This suggests that the 35.5–37 °C temperature range in human scrotal testes interrupts normal spermatogenesis, even though in elephants, this same temperature range maintains normal spermatogenesis. Furthermore, the koala has a similar body temperature as the elephant, although koalas perform spermatogenesis in the scrotum [35]. Although normal spermatogenesis occurs in the abdomen, poor semen quality in elephants collected manually has been reported [36,37], suggesting that the sperm maturation process in epididymis might be affected by abdominal temperature. Based on these results, we can exclude the possibility that normal spermatogenesis in elephant abdominal testes is due to a lower body temperature. Furthermore, cryptorchidism fails to suppress fertility in tropical rodents [38]. In tropical animals, including elephants, there may be a specific mechanism for producing sperm that is unaffected by heat. Because there is a limitation to knowing about the mechanism of elephant spermatogenesis under heat stress using the methods in the present study, further analysis from the different points of view will be needed to explore this research.

## 5. Conclusions

In conclusion, we identified two heat shock-induced mechanisms in elephant fibroblasts, consistent with other animals. One mechanism eliminates damaged cells through an HSF1/TDAG51-dependent apoptosis mechanism, whereas the other protects cells through an HSF1/HSP70-1-dependent mechanism. In elephant spermatogenic cells, there may be a mechanism to eliminate damaged cells through HSF1/TDAG51-dependent apoptosis; however, the protective effects of the HSF1/HSP70 dependent mechanism are not elucidated. The immunoexpression of HSP90 or HSP60 may support the heat stress tolerance in elephant spermatogenic cells. Therefore, a different protective mechanism is employed by spermatogenic cells to deal with heat stress, a mechanism that seems to be specific to elephants. However, it is not possible to non-invasively measure the temperature of the actual elephant testis; therefore, the effects of heat stress on spermatogenic cells need to be investigated in the near future using in vitro culture systems.

## Figures and Tables

**Figure 1 animals-14-02211-f001:**
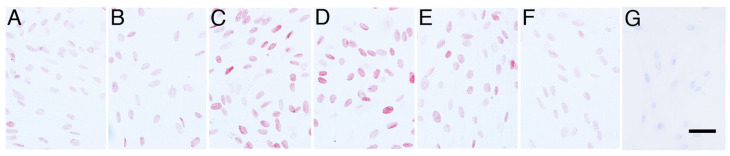
HSF1 immunoexpression in the primary cultures of elephant fibroblast cells after heat shock stimulus. (**A**) Cells without heat shock; (**B**) 0 h after heat shock; (**C**) 1 h after heat shock; (**D**) 2 h after heat shock; (**E**) 4 h after heat shock; (**F**) 6 h after heat shock; (**G**) negative control. The scale bar represents 50 µm.

**Figure 2 animals-14-02211-f002:**
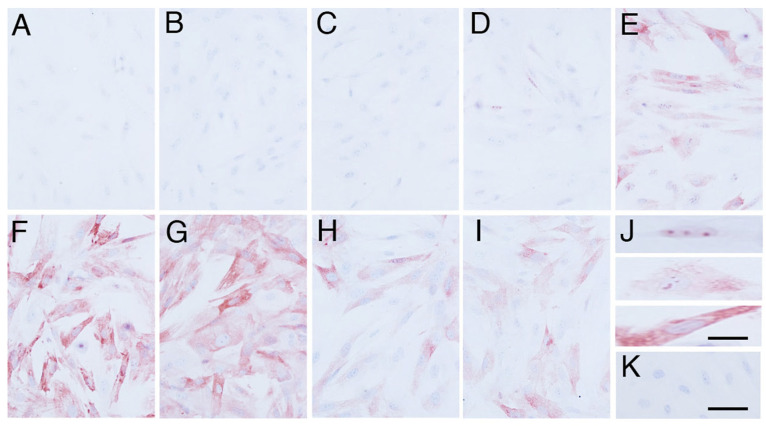
HSP70 immunolocalisation in the primary cultures of elephant fibroblast cells after heat shock stimulus. (**A**) Cells without heat shock; (**B**) 0 h after heat shock; (**C**) 1 h after heat shock; (**D**) 2 h after heat shock; (**E**) 4 h after heat shock; (**F**) 6 h after heat shock; (**G**) 12 h after heat shock; (**H**) 18 h after heat shock; (**I**) 24 h after heat shock; (**J**) localisation of HSP70 immunoexpression (upper; nucleus, middle; nucleus and cytoplasm, lower; cytoplasm); (**K**) negative control. The scale bar represents 50 µm (**A**–**I**,**K**), 20 µm (**J**).

**Figure 3 animals-14-02211-f003:**
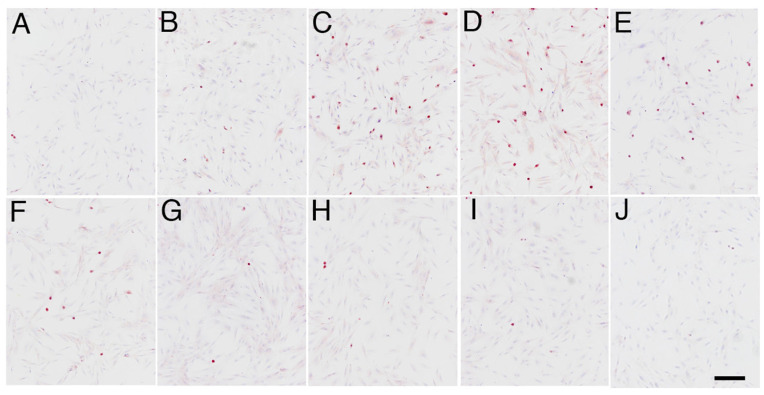
TDAG51 immunolocalisation in the primary cultures of elephant fibroblast cells after heat shock stimulus. (**A**) Cells without heat shock; (**B**) 0 h after heat shock; (**C**) 1 h after heat shock; (**D**) 2 h after heat shock; (**E**) 4 h after heat shock; (**F**) 6 h after heat shock; (**G**) 12 h after heat shock; (**H**) 18 h after heat shock; (**I**) 24 h after heat shock; (**J**) negative control. The scale bar represents 100 µm.

**Figure 4 animals-14-02211-f004:**
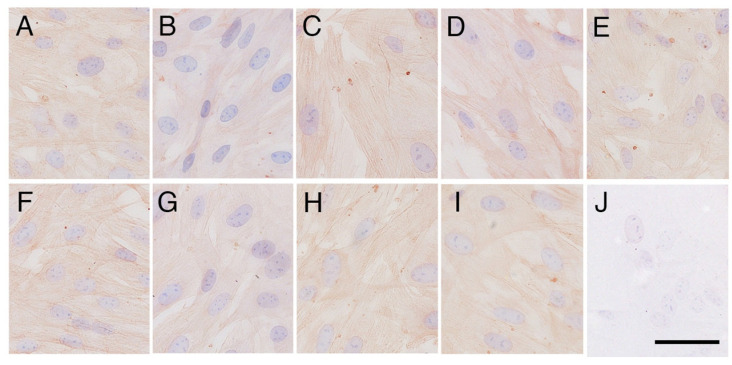
β-actin immunolocalisation in the primary cultures of elephant fibroblast cells after heat shock stimulus. (**A**) Cells without heat shock; (**B**) 0 h after heat shock; (**C**) 1 h after heat shock; (**D**) 2 h after heat shock; (**E**) 4 h after heat shock; (**F**) 6 h after heat shock; (**G**) 12 h after heat shock; (**H**) 18 h after heat shock; (**I**) 24 h after heat shock; (**J**) negative control. The scale bar represents 50 µm.

**Figure 5 animals-14-02211-f005:**
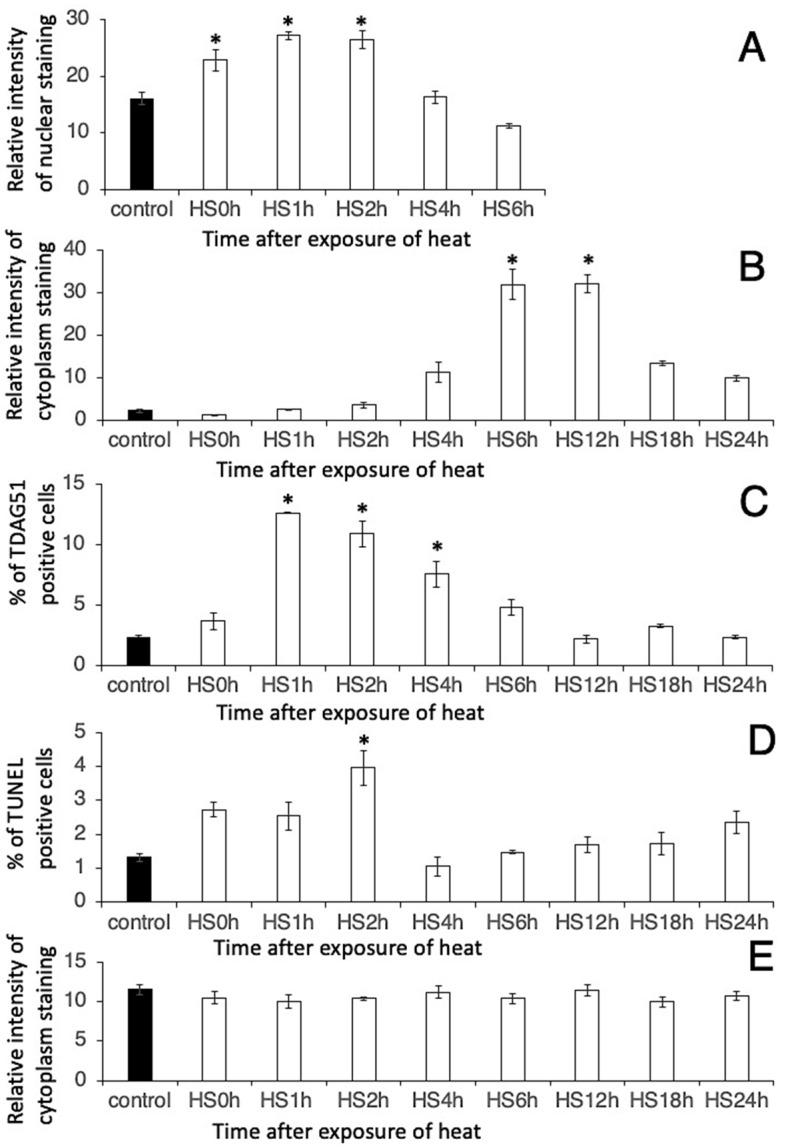
Semi-quantitative analysis of the immunoexpression of different proteins in the primary cultures of elephant cells after heat shock. (**A**) HSF1; (**B**) HSP70; (**C**) TDAG51; (**D**) TUNEL; (**E**) β-actin staining. Data are shown as a relative intensity of staining using an image analyser (**A**,**B**,**E**) and as the percentage of positive cells to total cells (**C**,**D**). The asterisk indicates the significant difference (*p* < 0.05) between heat shock stimulus (HS) and without heat shock stimulus (control); N = 4.

**Figure 6 animals-14-02211-f006:**
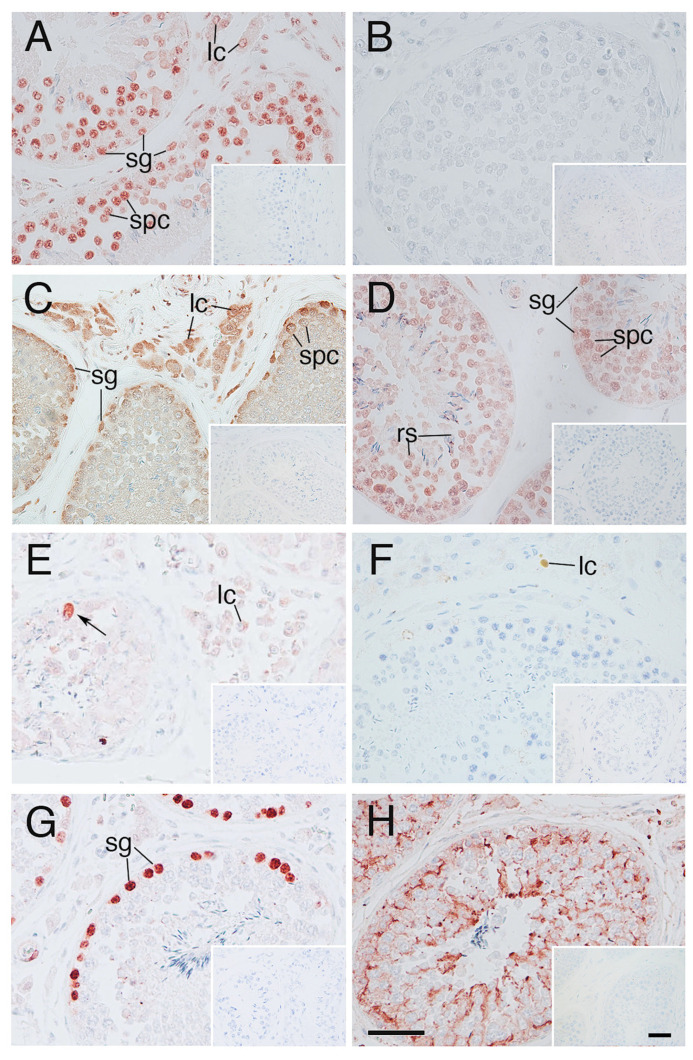
Immunohistochemistry of elephant tissues in the testes (**A**) HSF1; (**B**) HSP70; (**C**) HSP90; (**D**) HSP60; (**E**) TDAG51; (**F**) TUNEL; (**G**) PCNA; (**H**) β-actin. lc; Leydig cell, sg; spermatogonia, spc; spermatocyte, inset; negative control, Arrow; immunopositive spermatogenic cell, Scale bar; 50 µm.

**Table 1 animals-14-02211-t001:** Immunoexpression of each molecule in elephant fibroblast cells without heat shock.

	control	NH0h	NH1h	NH2h	NH4h	NH6h	NH12h	NH18h	NH24h
HSF1	9.38 ± 1.23	11.28 ± 1.43	11.06 ± 2.25	11.61 ± 1.02	11.15 ± 1.13	10.74 ± 0.87	9.27 ± 0.74		
HSP70	9.34 ± 0.44	11.61 ± 0.77	11.2 ± 0.31	10.76 ± 0.39	9.48 ± 0.43	10.08 ± 0.18	10.84 ± 0.40	9.49 ± 0.31	9.46 ± 0.28
TDAG51	1.39 ± 0.23	1.35 ± 0.11	1.15 ± 0.09	1.59 ± 0.16	1.12 ± 0.12	1.16 ± 0.11	1.53 ± 0.23	1.07 ± 0.09	1.61 ± 0.16
TUNEL	1.13 ± 0.14	1.10 ± 0.36	1.42 ± 0.11	0.60 ± 0.12	0.94 ± 0.11	0.81 ± 0.09	0.39 ± 0.10	0.56 ± 0.11	1.04 ± 0.14
b-act	13.76 ± 0.47	13.07 ± 1.51	13.42 ± 0.93	13.09 ± 1.70	9.51 ± 0.77	12.12 ± 0.95	11.91 ± 1.43	9.38 ± 0.90	10.90 ± 0.99

(Mean ± SE, N = 5); NH: without heat shock; HSF1: relative intensity of nuclear immunostaining, HSP70, b-act; relative intensity of nuclear immunostaining, TDAG51, TUNEL; % of immunopositive cells.

## Data Availability

The data that support the findings of this study are available from the corresponding author upon reasonable request.

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
