# Peer review of "Heat Shock Related Protein Expression in Abdominal Testes of Asian Elephant (Elephas maximus)"

_animals, 2024, doi:10.3390/ani14152211_

Round 1

Reviewer 1 Report

Comments and Suggestions for Authors

In their article, Sato et al. further explore the issue of spermatogenesis in the elephant abdominal testis. Regarding the phenomenon of testicular overheating, the authors studied changes in the immunoexpression of certain heat shock-related molecules. Due to the population of Asian elephants in some countries, this work may be of cognitive as well as practical importance (with regard to breeding these animals).

Nevertheless, the article needs some key corrections.

Line 37 - the term "expression" should only be used to refer to genes. For IHC staining, it is better to use the term "immuoexpression" or "immunolocalization".

Line 85, 99, 103 etc. - Testes are organs but not a tissue (from histological point of view). It means that there is no "testicular tissue".

Line 101 - without a doubt, the age of males is crucial in assessing spermatogenesis. Unfortunately, the current article does not say a word about the age of the animals.

Line 155 - unfortunately, the authors did not test the specificity of the antibodies used. The lack of any control (positive, negative or preadsorption tests) casts doubt on the obtained results. It is imperative that these issues be resolved in the revised version.

Line 198 - the statistical analysis is too enigmatic and incomplete. It lacks any information about normal distribution, post-hoc tests, homology of variance, etc.

Figure 5 - it is unclear what the lowercase letters actually mean?

Figure 6 - what do "lc", "sg" mean? "spc" spg" stand for? No explanation in the description to figure 6.

Line 396 - the authors should add limitations of the study. Their work is not comprehensive one so it is not difficult to find many such problematic issues (such as lack of rtPCR studies).

Author Response

We thank the reviewer for a detailed critique of our manuscript and also we thank the reviewer for understanding the importances of our work for reproduction in endangered species, Asian elephants. As much as possible, we made revise in line with the reviewer’s suggestions. We believe the reviewer’s valuable comments have significantly improved the overall quality of our work. We hope we have addressed all the comments adequately to be acceptable for publication in Animals.

Reviewer 2 Report

Comments and Suggestions for Authors

Animals-3111917

This study is designed to investigate the effects of “Heat shock related protein expression in abdominal testes of Asian elephant (Elephas maximus)”. The results suggest that elephant testicular cells have potential to eliminate damaged cells after heat shock stimulus through HSF1-TDAG51- mediated apoptosis. However, the protective and survival mechanisms differ from those in other mammals. The experimental design appears to be satisfactory and the manuscript is well-written. Nevertheless, it is recommended that the authors make revisions to the manuscript and address the following minor concerns:

Abstract:

the main objective and novelty of this study should be highlighted.

Introduction:

L67: environmental factors should be explained.

Material and methods:

L99-109: Sampling of elephant testicular tissues needs referencing along with more details in line with sampling from adult and calves.

L108-109: needs more details.

L126-155: ref should be added.

L181: camera?

L183: automatic image analyser?

L194: The examined fields were chosen randomly? Authors need explained in details

Results:

L205: needs to be revised.

Figure 5: Why are there big variation in SE between treatments.? For example, in A between HS 0h and HS 6h, in B between HS 6h with others, and etc.

In Figure 5: Comparisons between treatments needs to be re-analyzed, it seems there are no significant between some treatments and there are between others.

In Fig 5 and table 1, sample sizes are as a major concern.

Discussion:

L343: ........ a phenomenon consistent with other mammals (ref).

L348: ........ in the other mammals (such as ........ ?)

L365 and 366 should be continuous.

Comments on the Quality of English Language

Animals-3111917

This study is designed to investigate the effects of “Heat shock related protein expression in abdominal testes of Asian elephant (Elephas maximus)”. The results suggest that elephant testicular cells have potential to eliminate damaged cells after heat shock stimulus through HSF1-TDAG51- mediated apoptosis. However, the protective and survival mechanisms differ from those in other mammals. The experimental design appears to be satisfactory and the manuscript is well-written. Nevertheless, it is recommended that the authors make revisions to the manuscript and address the following minor concerns:

Abstract:

the main objective and novelty of this study should be highlighted.

Introduction:

L67: environmental factors should be explained.

Material and methods:

L99-109: Sampling of elephant testicular tissues needs referencing along with more details in line with sampling from adult and calves.

L108-109: needs more details.

L126-155: ref should be added.

L181: camera?

L183: automatic image analyser?

L194: The examined fields were chosen randomly? Authors need explained in details

Results:

L205: needs to be revised.

Figure 5: Why are there big variation in SE between treatments.? For example, in A between HS 0h and HS 6h, in B between HS 6h with others, and etc.

In Figure 5: Comparisons between treatments needs to be re-analyzed, it seems there are no significant between some treatments and there are between others.

In Fig 5 and table 1, sample sizes are as a major concern.

Discussion:

L343: ........ a phenomenon consistent with other mammals (ref).

L348: ........ in the other mammals (such as ........ ?)

L365 and 366 should be continuous.

Author Response

We thank the reviewer for a detailed critique of our manuscript. We also thank the reviewer for favorable comments to our manuscript. We are pleased to hear the reviewer found our manuscript well designed and written. We believe the reviewer’s valuable comments have significantly improved the overall quality of our work. We hope we have addressed all the comments adequately to be acceptable for publication in Animals.

Round 2

Reviewer 1 Report

Comments and Suggestions for Authors

 I accept explanations provided by the authors.